# Evaluation of Nano Fluids with Minimum Quantity Lubrication in Turning of Ni-Base Superalloy UDIMET 720

Onur Özbek 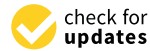

Department of Machine and Metal Technologies, Gumusova Vocational School, Duzce University, Duzce 81850, Turkey; onurozbek@duzce.edu.tr

**Abstract:** This article focuses on turning superalloy Udimet 720, which is difficult to work with, using different coolant/lubricant methods. The study includes delivering Graphene and Multi-Walled Carbon Nanotubes nanopowders homogeneously dispersed in vegetable oil to the cutting area with the minimum quantity lubrication (MQL) method. Experiments at different cutting speeds and feed rates were repeated in four different cutting environments. Compared to dry turning, the cutting zone temperature of the cutting fluid delivered to the cutting zone by MQL methods decreased. In addition, thanks to the nanopowders, it formed an oil film by better penetrating the cutting tool-chip interface and reducing the cutting tool's wear. With the reduced cutting tool wear, the cutting tool could maintain its form for a longer period of time, so better quality surfaces were obtained on the workpiece surface. As a result of the study, it was found that cutting zone temperature improved by 30%, tool wear by 51.8% and surface roughness by 43.9%.

**Keywords:** Ni-base superalloy; nano MQL; graphene; surface topography; cutting zone temperature; cutting tool wear

## 1. Introduction

Udimet 720 is a nickel-based superalloy with high frictional strength. It is produced by the powder metallurgy method. It is designed as the material of turbine discs used for nuclear gas-cooled reactors and other aerospace parts operating at high temperatures. Udimet 720, which can maintain its strength up to 1300 °C due to its low thermal conductivity, is very difficult to process. It is very difficult to obtain surface quality, which is important for the aviation and nuclear industry sectors [1,2]. The high temperature generated during chip removal is one of the most important aids in forming chips. However, due to their low thermal conductivity, superalloys retain their unique mechanical properties at high temperatures. This situation makes the machinability of superalloys very difficult. More force is required to remove chips from superalloys. As a result of this force, high temperatures occur in the cutting zone. However, this high temperature is absorbed by the cutting tool due to the low thermal conductivity of the workpiece. As it is known, these temperatures in the cutting zone accelerate the wear of the cutting tool. The surface roughness, in other words, the traces left behind by the chips broken off from the workpiece with which the wearing surfaces come into contact, are also affected by all these negativities. Therefore, machining superalloys poses a huge problem. For this reason, researchers have tried many methods to remove chips from materials with difficult machinability. Among these methods, cryogenic treatment [3,4], cryogenic machining [5–7], and MQL [8–11] are the ones that have attracted the most attention in the literature recently.

Due to the excessive use of oils for cooling and lubrication in machining, great harm is caused to the environment. In addition, operator health is adversely affected. For this reason, there has been an increasing awareness of green manufacturing recently. Researchers offer many new methods for green manufacturing to the use of industry. One of the newly developed methods, the MQL method, is one of them. In the MQL method, very little oil is

used in pulverized form, close to dry stock removal. Although the amount of oil used in the MQL method varies with other parameters, it is between 50–150 mL/h. The cutting oil film formed at the cutting tool-chip interface in the region where the cutting takes place causes many positive effects. Thus, better-quality surfaces can be obtained with MQL. In addition, the life of the cutting tools increases. Studies have reported that when vegetable oil is used as cutting oil, it has no negative effects on human health [12,13].

Research in the field of machinability has shown that cutting tool wear and surface roughness; it is affected by the cutting forces on the cutting tool [14], the vibration of the cutting tool [8], the power fluctuations of the machine [15], and the heat generated on the cutting tool [16]. The heat generated in the cutting zone can be quite high in superalloys. It has been reported in many studies that the cutting tool loses its hardness and wears rapidly at high temperatures. In the MQL method, the pulverized coolant sprayed to the tool-chip interface under high pressure reduces the cutting zone temperature [17,18]. Thus, cutting tool wear is reduced, and better surface roughness values are obtained. It also provides a great advantage in using much less oil than the traditional coolant method. Using vegetable oils that are completely biodegradable in the MQL method is a great advantage in terms of sustainability. Researchers have tried many vegetable oils for the MQL method. These vegetable oils include many oils such as coconut oil [19], jojoba oil [20], palm oil [13], sunflower oil [21], peanut oil [22], colza oil [23], and jatropha oil [24].

Recently, the unique properties of nanoparticles have been used in many areas. It has been reported that lower cutting tool wear is obtained by adding it to the cutting fluid in the MQL method. In these studies, it has been reported that nanoparticles improve the cutting fluid's thermal conductivity and penetration ability [25,26]. It has been reported that the friction force is reduced due to better penetration between the cutting tool and the workpiece [27]. Thus, lower temperatures occur between the cutting tool and the workpiece in the cutting zone. Cutting tool wear is reduced due to the lower cutting zone temperature. Due to reduced wear, the cutting tool maintains its form and can lift more stable workpieces from the contact surface. Thus, the surface roughness of the workpiece is directly affected, and the surface roughness values decrease. However, as in every field of machining, every parameter change has important results in the use of nanoparticles. Especially due to the chemical composition of the workpieces, the test results show significant differences. When the literature is examined, it is seen that the results vary according to the type, size, ratio, and solution preparation method of the nanoparticle used [28]. In these studies, it was observed that as the number of nanoparticles increased, the thermal conductivity and viscosity increased, and thus the penetration of the liquid decreased [29]. However, studies have shown that as the amount of nanoparticles increases, the viscosity increases, so the penetration of the liquid decreases.

In this article, nickel base super alloy nanoparticles, which are difficult to machine, were turned using the MQL method. The effects of nanoparticles on cutting zone temperature, cutting tool wear, and surface roughness were investigated.

## 2. Materials and Methods

The study used nickel-based super alloy Udimet 720 with Ø 152.6 × 15.55 mm dimensions as the workpiece material. The chemical composition of the turned workpiece is given in Table 1.

**Table 1.** Chemical components of Udimet 720 Nickel Base Alloy.

| Element | Cr | Co | Ti | Al | Mo | W | B | C | Ni |
|---|---|---|---|---|---|---|---|---|---|
| % | 16.38 | 14.84 | 5.74 | 2.7 | 2.55 | 1.27 | 0.02 | 0.02 | Bal. |

Experiments were carried out on the Accuway brand CNC lathe (Accuway Machinery, CO., Ltd., Taichung, Taiwan). TiAlN-TiN coated tungsten carbide cutting tools with the PVD method in the form of CNMG 120404 TT 5080 were used in the experiments. The experiments were carried out at three different feed rates (0.04, 0.06, 0.08 mm/rev),

three different cutting speeds (40, 60, 80 m/min), and a constant cutting depth (0.6 mm). All experiments were carried out at a constant chip volume of 2000 mm$^3$. Details of the experimental conditions are given in Table 2.

**Table 2.** Experimental conditions.

| Parameters | Level 1 | Level 2 | Level 3 | Level 4 |
|---|---|---|---|---|
| Feed rate (mm/rev) | 0.04 | 0.06 | 0.08 | - |
| Cutting speed (m/min) | 40 | 60 | 80 | - |
| Cutting conditions | Dry | Pure MQL | NanoMQL * (MWCNTs) | NanoMQL * (GNPs) |

* Multi-walled carbon nanotubes (MWCNTs); * Graphene Nano Platelets (GNPs).

The B1-210 model produced by Bielomatik (Cluj-Napoca, Romania) was used as the MQL system. Vegetable-based SAMNOS ZM-22W oil is used for sustainability as MQL oil. The oil used dissolves in nature at a rate of 100%, leaving no waste behind. The non-flammable MQL oil has a density of 1 g/cm$^3$ at 20 °C. With a flow rate of 150 (mL/h), the oil was delivered to the cutting area with 6 bar pressure from the nozzle of 1 mm diameter. Two different commercially available nanopowders and nanoparticle-added MQL cutting fluid were prepared for the experiments. Multi-walled carbon nanotubes and graphite nanoplatelets were used in MQL cutting fluids. The properties of the nanopowders used are given in Table 3.

**Table 3.** Multi-walled Carbon Nanotubes Doped with 12 wt% Graphene Nanopowder/Nanoparticles properties.

| MWCNT (Multi-Walled Carbon Nanotubes) | | GNP 12 wt% Graphene Nanopowder | |
|---|---|---|---|
| Purity | >97% | Purity | 99% |
| Avg. Inside Diameter (nm) | 5 | Thickness (nm) | 5 |
| Avg. Outside Diameter (nm) | >50 | Diameter (μm) | 1–12 |
| Specific Surface Area (m$^2$/g) | >65 | Specific Surface Area (m$^2$/g) | 500–1200 |
| Conductivity (s/m) | 1100–1600 | Conductivity (s/m) | 1000–1500 |
| Colour | black | Colour | grey |

The powders were added to the vegetable-based MQL oil at a rate of 0.5% by volume. Viscosities of prepared fresh MQL cutting fluids were measured with Fungilab, and pH values were measured with Orion Star A215 (Thermo Fisher Scientific Inc., Waltham, MA, USA). The obtained values are given in Table 4. Viscosity directly affects the thickness of the oil film that will form at the cutting tool-chip interface.

**Table 4.** MQL cutting fluids viscosity and pH values.

| | Pure MQL | nMQL GNP | nMQL MWCNT |
|---|---|---|---|
| pH | 7.75 | 7.89 | 8.02 |
| Viscosity (mPa.s) | 5 | 6.3 | 6.5 |

MQL cutting fluid was used fresh after three different mixing processes. MQL cutting fluid was mixed in HeidolpH–Hei-Torque Precision 200 mechanical stirrer (Heidolph Instruments, Schwabach, Germany), Bandelin Sonopuls–UW 3200 ultrasonic stirrer (Bandelin, Mecklenburg-Vorpommern, Germany), and finally, Thermal–N11150M magnetic stirrer (THERMAL, Minden, NV, USA) for 60 min, respectively. All cutting parameters were tested in four different cutting conditions. Vegetable-based Nano MQL cutting fluid production process is given in Figure 1.

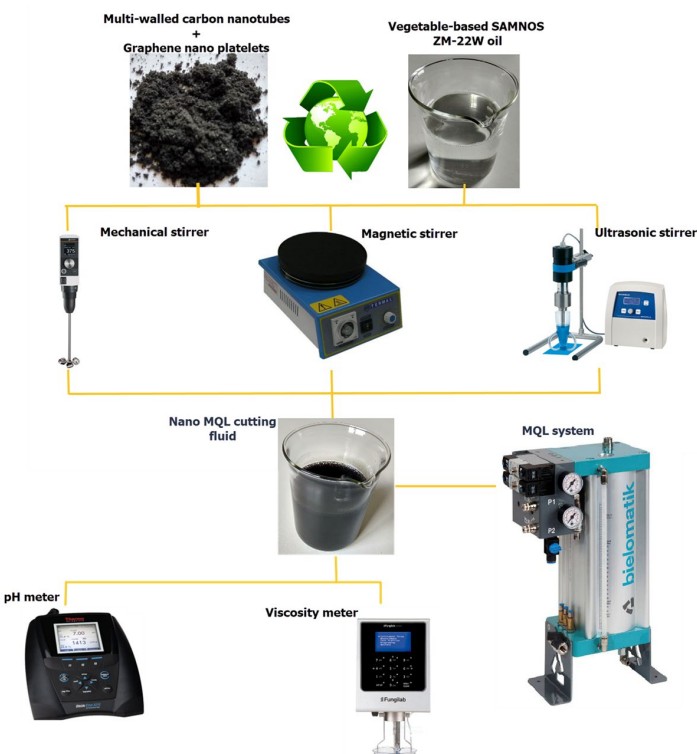

**Figure 1.** Vegetable-based Nano MQL cutting fluid production process.

During all experiments, the temperature in the cutting area determined at the cutting tool tip was measured with the Optris PI 450 thermal camera (Optris, Berlin, Germany). The wear images of the cutting tools were taken at the macro-scale with DINO LITE 2.0 microscope (DINO LITE, Torrance, CA, USA) and at the micro-scale with FEI Quanta FEG 250 (FEI, Hillsboro, OR, USA) scanning electron microscope (SEM). The surface quality formed after chip removal was measured with a Mahr PS10 profilometer (Mahr, Providence, RI, USA). Both Ra and Rz measurements of the surface were performed. Phase View optical profilometer was used to examine the topography of the processed surface. The experimental setup is given in Figure 2.

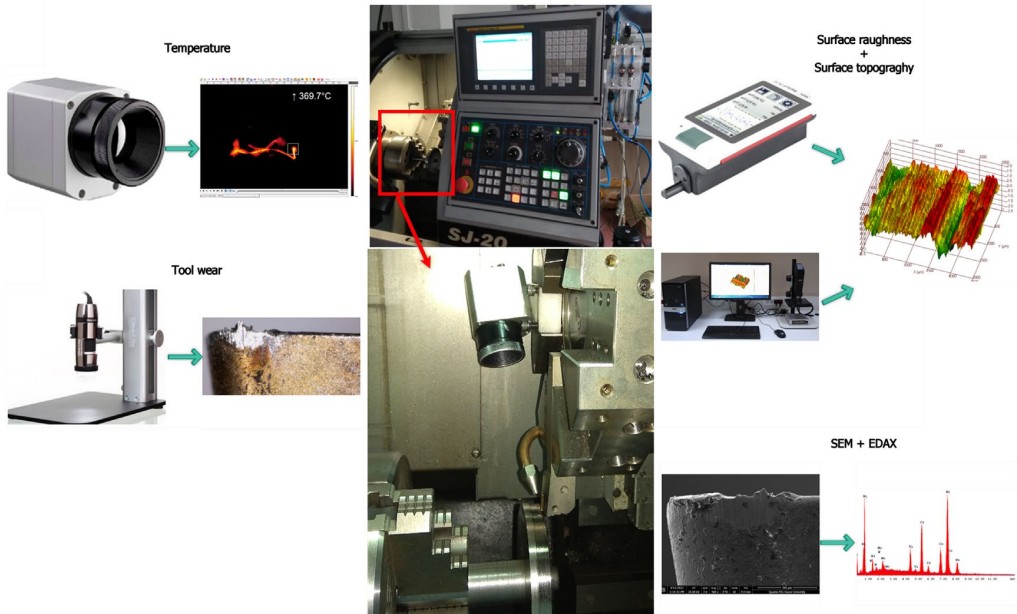

**Figure 2.** Experimental setup.

## 3. Results and Discussion

### 3.1. Cutting Temperature

High temperature occurs when two metals with high strength are rubbed. The high temperature formed in the cutting zone changes the tribological properties of the cutting tool [30]. For this reason, the hardness of the cutting tool decreases and causes faster wear [8]. The rapidly wearing cutting tool also completes its life in a shorter time. In addition, the surface quality of the workpiece deteriorates at a similar rate. Superalloys maintain their yield strength at high temperatures and are difficult to process. Therefore, the cutting zone temperature of superalloys is also high. The thermal conductivity of the Ni-based superalloy Udimet 720 is about 20 W/mK, while the thermal conductivity of normal steel is approximately 50 W/mK. [31]. This difference in thermal conductivity shows that the temperature formed in the cutting zone is less away with the workpiece or chip. In normal steel materials, most of the temperature formed in the cutting zone is removed from the area by the workpiece and chip. However, in superalloys, the cutting tool absorbs the temperature that the workpiece or chip cannot remove. The cutting tool, which has more thermal load, is also expected to wear faster. In addition, due to the worn cutting tool, the workpiece's dimensional accuracy and the workpiece's surface quality deteriorate. In order to measure the actual temperature in the cutting zone, the area where the temperature data is taken was determined as 5 mm$^2$.

Figure 3 shows the cutting zone temperatures taken from the thermal camera at the highest feed and cutting speed. Significant effects of MQL liquid and nanoparticles delivered to the cutting zone on the cutting temperature were determined.

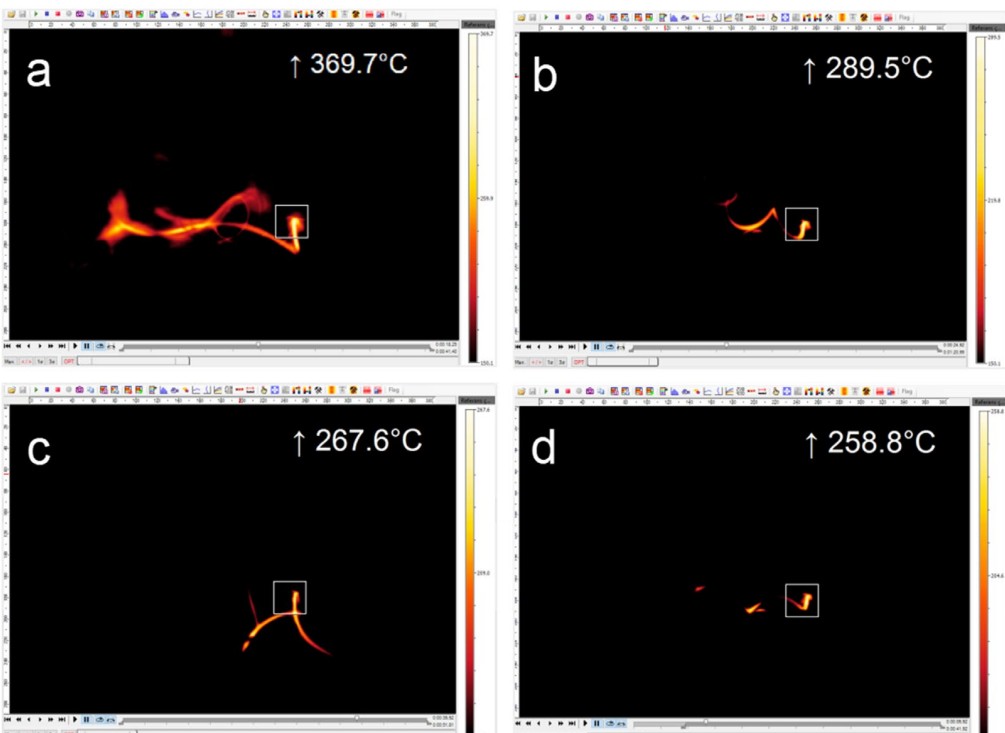

**Figure 3.** Thermal picture of Ni-based superalloy turning under different conditions at a cutting speed of 80 m/min and a feed rate of 0.08 mm/rev: (**a**) Dry; (**b**) Pure MQL; (**c**) nMQL GNP; (**d**) nMQL MWCNT.

It is seen in Figure 4 that the cutting zone temperature increases with the increase of cutting speed and feed rate in all cutting conditions. When the cutting zone temperature is examined according to the cutting parameters, the cutting zone temperature decreases by 19.53%, decreasing the cutting speed from 80 m/min to 40 m/min in dry turning. The cutting zone temperature decreased by approximately 10.39% as the feed rate decreased

from 0.08 mm/rev to 0.04 mm/rev. The fact that the cutting speed affects almost twice as proportionally to the feed rate proves how important friction is in superalloys. With the increase in feed rate, the power consumed for chip removal turns into heat energy, while the friction increases with the increase in cutting speed. In this case, it can be said that the energy released from friction is more. However, the actual temperature difference was observed in the cutting conditions changes. When the cutting zone temperature was evaluated according to the cutting conditions, Pure MQL reduced the cutting zone temperature by 21.7% compared to dry turning. This rate increased to approximately 27.6% when nMQL GNP was used. The cutting condition with the lowest cutting zone temperature was nMQL MWCNT, which showed a 30% decrease compared to dry turning. When MQL coolant reaches the high-temperature cutting zone at room temperature, the cutting zone temperature is expected to decrease. Thus, less heat is generated in the cutting zone. Nanoparticles added to the MQL fluid increase the thermal conductivity of the cutting fluid. Therefore, it causes the heat collected at the cutting tool tip to move away from this region faster. In addition, it is thought that nanoparticles increase the lubricating properties of the cutting fluid, reducing friction. The SEM images obtained as a result of the experiments also prove that nanoparticles reduce the formation of BUE. BUE formation in machining has many known disadvantages. The most important of these is the deterioration of the cutting tool form and the increase in the cutting zone temperature.

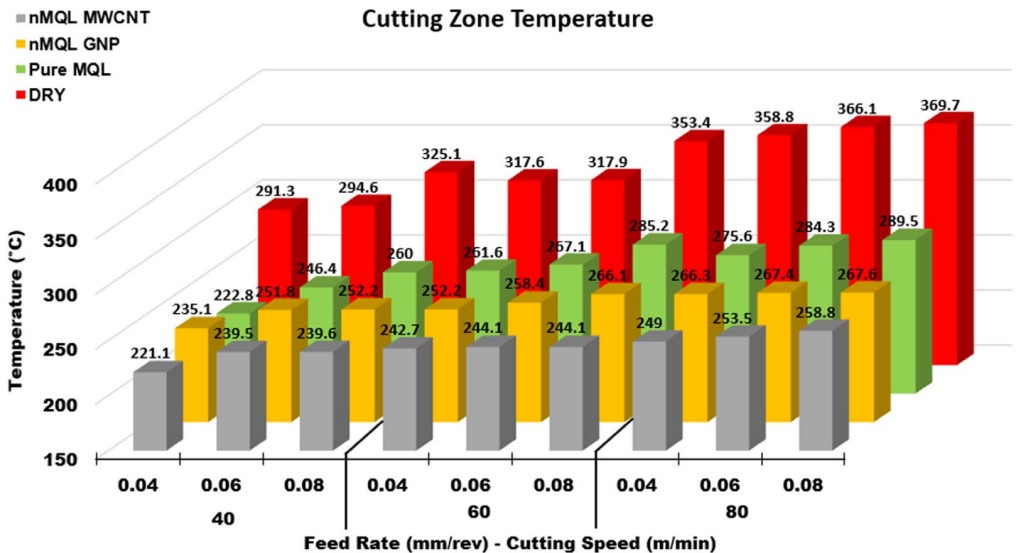

**Figure 4.** Cutting zone temperature variation.

### 3.2. Flank Wear

The high thermal conductivity of nanofluids maximizes penetration in the cutting zone, thus reducing friction, cutting zone temperature, and cutting tool wear [32]. In Figure 5, at a constant chip volume of 2000 mm$^3$, the maximum flank wear in dry turning was 0.185 mm at the highest cutting parameters. The lowest flank wear was measured as 0.052 mm, decreasing by 71.89% when using the lowest cutting parameters with nMQL MWCNT. In dry turning, cutting tool wear increased with an increase in cutting speed at a feed rate of 0.4 mm/rev. By increasing the cutting speed by 100%, the wear of the cutting tool increased by 36.84%. While the feed rate was 0.6 mm/rev, the wear of the cutting tool increased by 32.22% with the increase in cutting speed. When the feed rate was 0.8 mm/rev, the wear of the cutting tool increased by 27.56% with the increase in the cutting speed. With the feed rate change, the cutting zone temperature increased by 10.39% at most. When the effects of different cutting media on wear were examined, significant differences were observed at the highest cutting speed and feed rates. When using the Pure MQL method compared to dry turning, the wear of the cutting tool decreased by 9.19%. This rate was 14.05% when nMQL GNP was used, while nMQL MWCNT improved

by 15.67%. The differences in the cutting environment are quite remarkable in the lowest cutting parameters that can be selected in cases where tool wear is important. The cutting tool wear, measured as 0.108 mm in dry turning, decreased by 35.1% with Pure MQL and was measured as 0.07 mm. While this rate was 39.8% when nMQL GNP was used, 51.8% improvement was observed for nMQL MWCNT. nMQL MWCNT can be considered a proven option, especially when it is desired to have low surface roughness values, which can be considered as a result of cutting tool wear. Another factor that needs attention is the correct selection of cutting parameters. No catastrophic failure was observed, even in dry turning at the highest feed and cutting speeds. Moreover, the 0.2 mm wear limit was complied with, where the tool was considered worn and unusable.

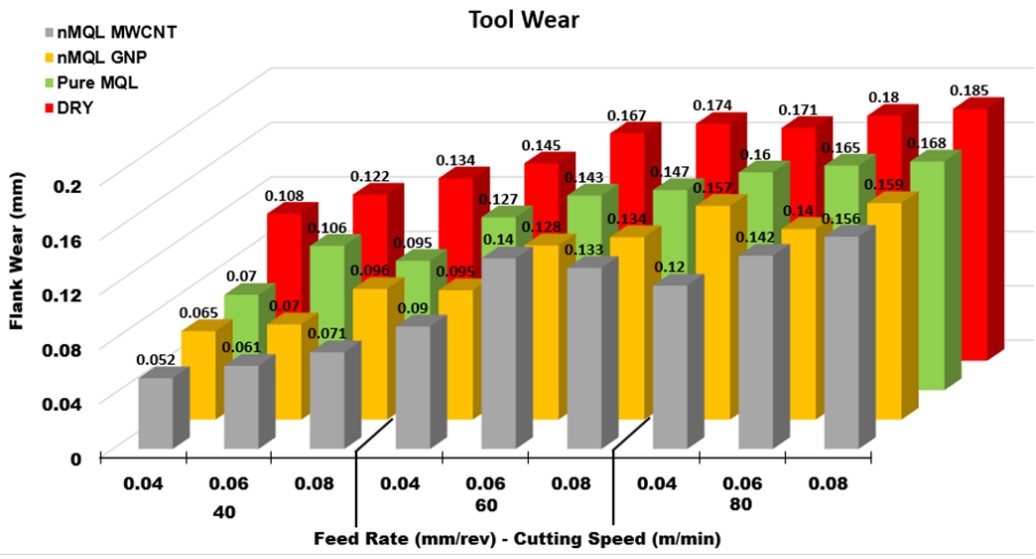

**Figure 5.** Change of flank wear.

After all the experiments, macro images of the cutting tools were taken under DINO LITE 2.0 optical microscope. Table 5 shows the wearing surfaces of all cutting tools. While BUE formation is observed especially in dry turning, a decrease is observed when Pure MQL is used. In addition, with the effect of nanopowders, BUE formation was considerably reduced. This situation is proved by the SEM images in Figure 6. The images also appear to remove the coating that nMQL GNP failed to protect the cutting tool coating. This situation is thought to cause increased wear.

**Table 5.** Images of cutting tool wear.

| Feed Rate, mm/rev | Cutting Speed, m/min | | |
|---|---|---|---|
| | 40 | 60 | 80 |
| Dry    0.04 | | | |
| 0.06 | | | |
| 0.08 | | | |

**Table 5.** *Cont.*

| Feed Rate, mm/rev | Cutting Speed, m/min | | |
|---|---|---|---|
| | 40 | 60 | 80 |

**Pure MQL**

| | 0.04 | | |
|---|---|---|---|
| | 0.06 | | |
| | 0.08 | | |

**nMQL GNP**

| | 0.04 | | |
|---|---|---|---|
| | 0.06 | | |
| | 0.08 | | |

**nMQL MWCNT**

| | 0.04 | | |
|---|---|---|---|
| | 0.06 | | |
| | 0.08 | | |

With the effect of high temperature in the cutting zone, the cutting tool loses its hardness. This causes rapid wear of the cutting tool. In the SEM images seen in Figure 5, it is seen that flank wear is higher in Dry with a high cutting zone temperature. In addition, EDS analysis proves the BUE formed on the cutting tool. EDS results show that BUE formation is very high with the effect of heat, especially in dry turning. With Pure MQL, the BUE formation is slightly reduced, but the intensity of the wear is noticeable. BUE does not appear when using nMQL GNP. However, this does not mean that BUE does not occur. It is understood from the SEM and EDX no. 3 that the BUE formed, broke off, and took a part of the coating with it when breaking off. When the SEM image no. 3 is examined, and the W element, which is the cutting tool substrate, is seen instead of the remains from the TiN or TiAlN layers, which are the upper layer of the cutting tool. Removal of the coating is an important disadvantage. It is known that when BUE moves away from the environment, it moves away with the coating. Therefore, an important wear problem can be mentioned here due to BUE. However, BUE and wear were observed to be quite low when nMQL

MWCNT was used. It can be said that nMQL MWCNT causes low wear and low BUE formation due to its low cutting temperature. This also affected the wear results.

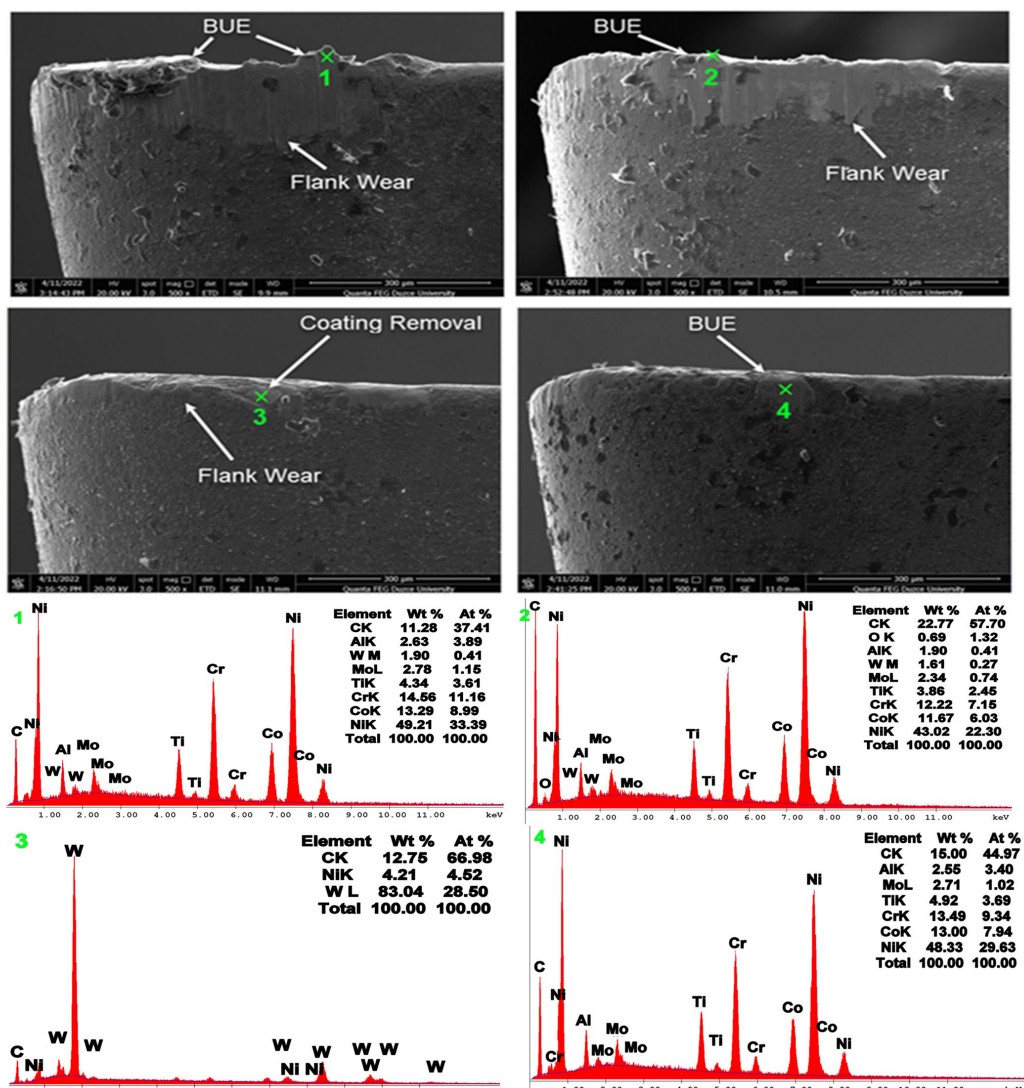

**Figure 6.** SEM images and EDS analysis of the cutting tool.

### 3.3. Surface Roughness and Surface Topography

The surface roughness is expected to increase with the increase in friction between the cutting tool and the workpiece [33]. Increasing the feed rate and cutting speed increases the cutting forces and, thus, the friction between the cutting tool and the workpiece [16]. As expected, a noticeable increase was observed in the surface roughness values with the increase in feed rate and cutting speed (Figure 7). However, it is clearly seen that MQL and nanoparticle-added MQL have the main effect on surface roughness rather than cutting parameters. When the surface roughness values after dry turning were examined, the surface roughness improved by 12.3% and 36.5%, respectively, with a 100% decrease in feed rate and cutting speed. With Pure MQL, the surface roughness value improved by 21.4%. The cutting temperature decreased with the MQL fluid reaching this region, which positively affected tool wear. The cutting tool, which wears less and maintains its form with MQL coolant, resulted in better surface quality with the reduction of friction with MQL coolant. It has been reported in previous studies that the mechanical and thermal load on the workpiece decreases when nanoparticles are added to the cutting fluid [32,34]. The reason for this is the high thermal conductivity of nanofluids and their ability to lower surface tension. When using nMQL GNP as nanopowder, the surface roughness improves

by 37.5% compared to dry turning, while this rate increases up to 43.9% when nMQL MWCNT is used. nMQL, which gives the best result in surface roughness compared to dry turning, provided an 88% improvement by reducing the MWCNT cutting parameters by 100%.

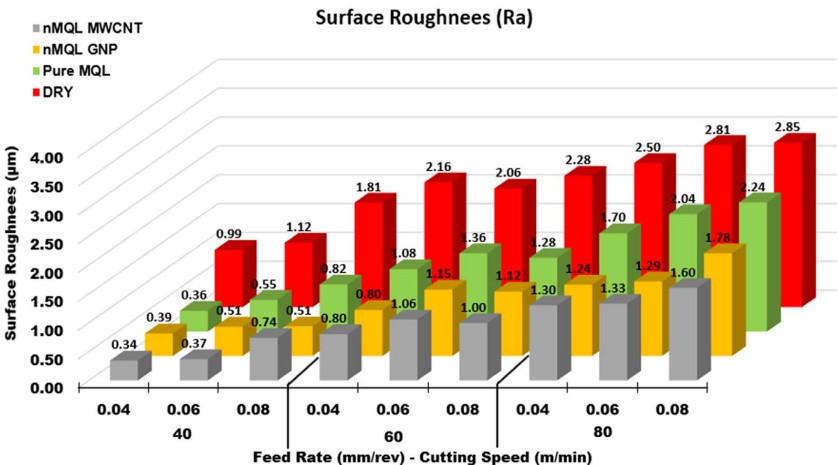

**Figure 7.** Change of surface roughness.

In Figure 8, the surface topographies obtained from the turning experiments performed at different cutting conditions (DRY, Pure MQL, nMQL GNP, and nMQL MWCNT) at cutting speed (60 m/min) and feed rate (0.06 mm/rev) are given. Topographic images obtained from the workpiece show that the use of nanopowders significantly improves surface roughness.

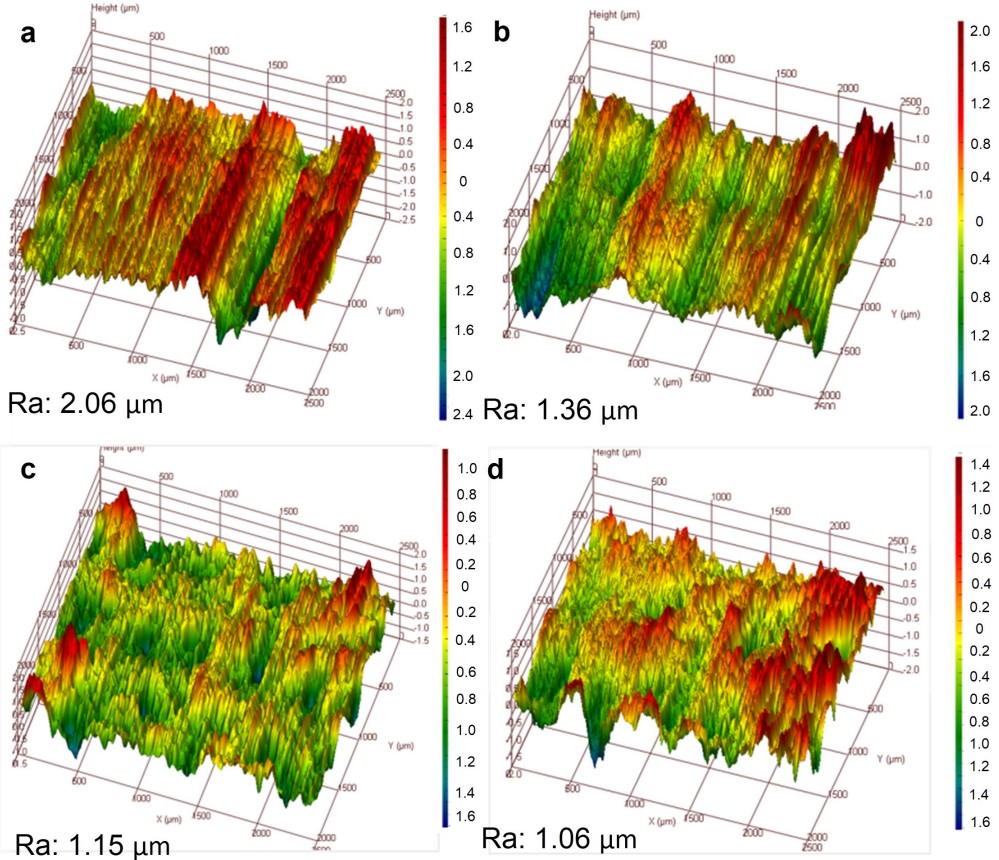

**Figure 8.** Change of surface topography (cutting speed of 60 m/min and a feed rate of 0.06 mm/rev). (**a**) Dry; (**b**) Pure MQL; (**c**) nMQL GNP; (**d**) nMQL MWCNT.

## 4. Conclusions

This paper investigated three MQL methods as an alternative to dry turning of superalloy Udimet 720, which is difficult to machine. These are (i) pure MQL, (ii) nMQL GNP, and (iii) nMQL MWCNT.

In this study, in which the effect of cutting parameters was also investigated, the experiments were carried out at three different feed rates (0.04, 0.06, 0.08 mm/rev), three different cutting speeds (40, 60, 80 m/min), and a fixed depth of cut (0.6 mm) for all conditions.

With the cutting speed change, the highest temperature difference was 19.53%. With the feed rate change, the cutting zone temperature increased by 10.39% at most. This shows that the cutting speed is more effective than the feed rate at the cutting zone temperature. Based on the highest cutting speed and feed rate values, the cutting zone temperature decreased by 21.7%, 27.6%, and 30% in Pure MQL, GNP, and MWCNT, respectively, compared to dry turning.

In all cutting environments, the cutting tool wear reached its maximum at a higher cutting speed and feed rate. With the reduction of cutting speed and feed rate in dry turning, cutting tool wear has also decreased by approximately 36% and up to 27%.

The highest cutting tool wear occurred in the dry turning environment. When Pure MQL, nMQL GNP, and nMQL MWCNT were used at the lowest cutting parameters, cutting tool wear was reduced by approximately 35%, 39%, and 51%, respectively, compared to dry turning. At the highest cutting parameters, this reduction in cutting tool wear was approximately 9%, 14%, and 15%, respectively.

Higher cutting speed and feed rate resulted in higher surface roughness. In dry turning, an improvement of approximately 36% and up to 12% was observed in the surface roughness by reducing the cutting speed and feed rate.

The highest surface roughness occurred in the dry-turning environment. Using Pure MQL, nMQL GNP, and nMQL MWCNT at the highest cutting parameters resulted in approximately 21%, 37%, and 43% lower surface roughness, respectively, compared to dry turning. Compared to dry turning, the most significant improvement in surface roughness was achieved when nMQL MWCNT was used, with a rate of approximately 88%.

Studies can be carried out with mathematical models as a contribution to the current research. In addition, many nanoparticles are added to cutting fluids, and their effects are tested. Experiments with different particles can be performed in Udimet 720.

**Funding:** This research received no external funding.

**Data Availability Statement:** Data are contained within the article.

**Conflicts of Interest:** The authors declare no conflict of interest.

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
