# Peer review of "Evaluation of Nano Fluids with Minimum Quantity Lubrication in Turning of Ni-Base Superalloy UDIMET 720"

_lubricants, doi:10.3390/lubricants11040159_

Round 1

Reviewer 1 Report

Decision: Minor Revision

The authors have studied “Evaluation of nano fluids with minimum quantity lubrication in turning of Ni-base superalloy UDIMET 720”. It’s interesting article and have many applications in the working nanofluid dynamics. The presentation of the article is very good and it falls under the aims and scope of the journal.

Comments to the Authors:

1. There are many grammatical or spelling issues in the manuscript, which can make reading difficult. Add proper punctuations to the text as well as to the equations.

2.    Give references to the equations those authors did not derived.

3.The updated literature reference seems to be limited and it should encompass the current level of knowledge, Specifically

·         Experimental study of improved Rheology and Lubricity of drilling fluid enhanced with nano-particles, Applied Nanoscience, Vol. 8(5), pp. 1069-1090, 2018.

4.   Future works with applicability should be included at the end.

Author Response

The authors have studied “Evaluation of nano fluids with minimum quantity lubrication in turning of Ni-base superalloy UDIMET 720”. It’s interesting article and have many applications in the working nanofluid dynamics. The presentation of the article is very good and it falls under the aims and scope of the journal.

Response:

Thank you for your nice thoughts on the article.

  1. There are many grammatical or spelling issues in the manuscript, which can make reading difficult. Add proper punctuations to the text as well as to the equations.

Response:

Appropriate punctuation marks have been added to the text.

  1. Give references to the equations those authors did not derived.

Response:

Thank you very much for your comment. The article has been revised.

3.The updated literature reference seems to be limited and it should encompass the current level of knowledge, Specifically

  • Experimental study of improved Rheology and Lubricity of drilling fluid enhanced with nano-particles, Applied Nanoscience, Vol. 8(5), pp. 1069-1090, 2018.

Response:

Thank you very much for your valuable opinion regarding the information provided. The article you requested has been added to the literature.

  1. Future works with applicability should be included at the end.

Response:

Thank you very much for your comment. Studies with applicability are included at the end of the article.

Reviewer 2 Report

Evaluation of nano fluids with minimum quantity lubrication in turning of Ni-base superalloy UDIMET 720 

Good topic to discuss MQL in machining operations.

Operative temperature is always a problem for quality of working pieces and also for tooling useful life. Section 3.1 address this topic in detail.

A schematic of the tool/working piece of the system used would be helpful for readers to understan what author refer to machining operation.

Image 5 with the cutting tool wear is good to comprehend the effects of machining parameters on the tool wear.

Conclusions > Different scenarios and parameters play critical roles, which were addressed in this manuscript.

Zone temperature improved by 30%, tool wear >50% and surface roughness improved by more than 40%

Author Response

Reviewer 2

Evaluation of nano fluids with minimum quantity lubrication in turning of Ni-base superalloy UDIMET 720 

Good topic to discuss MQL in machining operations.

Operative temperature is always a problem for quality of working pieces and also for tooling useful life. Section 3.1 address this topic in detail.

Response:

Thank you very much for your valuable opinion regarding the information provided. Cutting temperature is discussed in detail in Section 3.1. The cutting zone temperature was measured with a thermal camera, and the data was interpreted.

A schematic of the tool/working piece of the system used would be helpful for readers to understan what author refer to machining operation.

Response:

A graphic summary has been added to the article.

Image 5 with the cutting tool wear is good to comprehend the effects of machining parameters on the tool wear.

Response:

Thank you very much for your comment.

Conclusions > Different scenarios and parameters play critical roles, which were addressed in this manuscript.

Zone temperature improved by 30%, tool wear >50% and surface roughness improved by more than 40%

Response:

Thank you very much for your comment. It has been detailed since it was requested to detail the results and to write the reasons in the previous control. The conclusions have been revised.

Reviewer 3 Report

There are very few questions and minor corrections:

Why is the difference between the values of viscosity of MQL oil presented in line 107, respectively Table 4 (it is not from measurement unit)?

Figure 3: The diagram shows like you made 3 different samples of oil with different mixing processes. I think the mixing processes are made in a subsequent order for each oil sample (in series not in parallel)

Figures 4, 5 and 7: Feed rate comes first on the axis and then cutting speed, not “Cutting speed – Feed rate”

Timing and order of experiments, also some results are not clear:

Do you use a new cutting tool for each constant cutting speed, for each speed rate and each condition of lubrication?

How is the temperature changing in time? Is the measured temperature stabilized? After which period of time from starting cutting is it measured? Are the parts completely cooled before restarting the measurements?

How do you measure wear? At which point of flank, in which direction, relative to what?

The Conclusions section should not repeat detailed information from the previous sections.

Author Response

Reviewer 3

There are very few questions and minor corrections:

Why is the difference between the values of viscosity of MQL oil presented in line 107, respectively Table 4 (it is not from measurement unit)?

Response:

Thank you very much for your comment. The values given in Table 4 are dynamic viscosity values. I measured these values with a Fungilab brand device. These values are left in the article. The other value is the value given by the manufacturer. This value is the kinematic viscosity value. The kinematic viscosity value has been omitted from the article to avoid confusion. Measurements were made with the same method for all prepared cutting fluids.

Figure 3: The diagram shows like you made 3 different samples of oil with different mixing processes. I think the mixing processes are made in a subsequent order for each oil sample (in series not in parallel)

Figures 4, 5 and 7: Feed rate comes first on the axis and then cutting speed, not “Cutting speed – Feed rate”

Response:

Thank you very much for your comment. Three different mixing processes were used for both oil samples with nanoparticles added. Mechanical, ultrasonic and magnetic stirrers mixed each sample. Thus, the nanoparticle-added MQL cutting fluid became more homogeneous. Changes have been made for Figs 4.5 and 7.

Timing and order of experiments, also some results are not clear:

Do you use a new cutting tool for each constant cutting speed, for each speed rate and each condition of lubrication?

Response:

Thank you very much for your comment. A new cutting tool was used for each different experiment and its repetition. Thus, both cutting parameters and different conditions of lubrication were seen.

How is the temperature changing in time? Is the measured temperature stabilized? After which period of time from starting cutting is it measured? Are the parts completely cooled before restarting the measurements?

Response:

Cutting tool wear was measured after each test. Surface roughness measurement was carried out on the workpiece. A new cutting tool was used for all experiments. Sufficient time was allowed for the workpiece to cool after each experiment. Defined as the cutting zone with the thermal camera program "The area where the temperature data is taken to measure the actual temperature in the cutting zone is 5 mm2." Max. temperature was taken.

How do you measure wear? At which point of flank, in which direction, relative to what?

Response:

Thank you very much for your comment. The article focuses on flank wear, which is seen as tool wear. The cutting tool nose diameter was measured after 0.8 mm. "The wear images of the cutting 130 tools were taken at the macro-scale with DINO LITE 2.0 microscope and at the micro-scale 131 with FEI Quanta FEG 250 scanning electron microscope (SEM)."

The Conclusions section should not repeat detailed information from the previous sections.

Response:

Thank you very much for your comment. The conclusions have been revised.
